# Mutation-Based Therapeutic Strategies for Duchenne Muscular Dystrophy: From Genetic Diagnosis to Therapy

**DOI:** 10.3390/jpm9010016

**Published:** 2019-03-04

**Authors:** Akinori Nakamura

**Affiliations:** 1Department of Neurology, National Hospital Organization, Matsumoto Medical Center, 2-20-30 Murai-machi Minami, Matsumoto 399-8701, Japan; anakamu@shinshu-u.ac.jp; Tel.: +81-263-58-4567; 2Third Department of Medicine, Shinshu University School of Medicine, 3-1-1 Asahi, Matsumoto 390-8621, Japan

**Keywords:** Duchenne/Becker muscular dystrophy (DMD/BMD), *DMD* gene, dystrophin, molecular diagnosis, DMD model animal, gene therapy, readthrough therapy, exon skipping therapy, antisense oligonucleotide

## Abstract

Duchenne and Becker muscular dystrophy (DMD/BMD) are X-linked muscle disorders caused by mutations of the *DMD* gene, which encodes the subsarcolemmal protein dystrophin. In DMD, dystrophin is not expressed due to a disruption in the reading frame of the *DMD* gene, resulting in a severe phenotype. Becker muscular dystrophy exhibits a milder phenotype, having mutations that maintain the reading frame and allow for the production of truncated dystrophin. To date, various therapeutic approaches for DMD have been extensively developed. However, the pathomechanism is quite complex despite it being a single gene disorder, and dystrophin is expressed not only in a large amount of skeletal muscle but also in cardiac, vascular, intestinal smooth muscle, and nervous system tissue. Thus, the most appropriate therapy would be complementation or restoration of dystrophin expression, such as gene therapy using viral vectors, readthrough therapy, or exon skipping therapy. Among them, exon skipping therapy with antisense oligonucleotides can restore the reading frame and yield the conversion of a severe phenotype to one that is mild. In this paper, I present the significance of molecular diagnosis and the development of mutation-based therapeutic strategies to complement or restore dystrophin expression.

## 1. Introduction

The dystrophinopathies Duchenne muscular dystrophy (DMD) and Becker muscular dystrophy (BMD) are X-linked muscle diseases caused by a lack of the subsarcolemmal protein dystrophin due to mutations in the dystrophin (*DMD*) gene [1,2]. Duchenne muscular dystrophy is typically diagnosed between two and five years of age and presents with an abnormal gait and whole-body skeletal muscle wasting, which progress due to a marked increase in serum creatine kinase (CK), typically resulting in the inability to walk before the age of 16 years [3,4]. These patients often die around 30 years of age due to respiratory or heart failure [3,4]. The pooled prevalence rate of DMD is 4.78 per 100,000 males and incidence ranges from 10.71 to 27.78 per 100,000 males [5]. In general, DMD is first perceived as a retardation of motor development, but family history or an unexpected discovery of high serum CK levels in blood tests allows for the diagnosis of this disease even when it is still in the early, asymptomatic stage. Genetic counseling is difficult because approximately 1/3 of the patients have DMD due to a de novo mutation.

Becker muscular dystrophy (BMD) is a milder variant of DMD and also shows motor development retardation, hyperCKemia, calf hypertrophy, and dilated cardiomyopathy (DCM). Its severity markedly varies among patients, ranging from asymptomatic to early death due to cardiac failure [6,7,8,9]. The incidence of BMD is 1.53 per 100,000 male births [5].

## 2. The *DMD* Gene and its Mutations in Dystrophinopathy

Located at Xp21, the *DMD* gene has 79 exons spanning ~2500 kb in genomic DNA and encodes the subsarcolemmal protein dystrophin [1,2]. mRNA of the *DMD* gene is mainly expressed in skeletal, cardiac, and smooth muscles as well as in the brain. Dystrophinopathy is diagnosed by identifying the *DMD* gene mutation, frequently via multiplex ligation-dependent probe amplification (MLPA), which quickly and accurately identifies mutations in ~70% of dystrophinopathy cases that have deletion(s) and/or duplication(s) of the 79 exons in the *DMD* gene [10]. In dystrophinopathy, *DMD* gene mutations include deletions of one or more exons (60–70% of cases) [11,12,13,14] and duplications (5–10%) [14,15,16,17] (Table 1). The phenotypic distinction between DMD and BMD is mostly explained by the reading frame theory [18,19]. When the reading-frame of amino acids is disrupted by a mutation (out-of-frame), dystrophin is not expressed, resulting in the severe DMD phenotype [4,5]. Contrarily, maintaining the reading frame despite a mutation (in-frame), results in dystrophin production, leading to a milder phenotype, such as BMD. The BMD phenotype also varies because of the different structure, function, and expression level of truncated dystrophin, which arises due to mutation type, size, and location within the *DMD* gene [18,19,20]. This frame-shift theory illustrates the differences in phenotype of ~90% of dystrophinopathies [15,18,21,22]. There are two genetic hotspots of deletion mutations in the *DMD* gene, exons 45–55 and exons 3–7 [10]; these regions account for approximately 60% and 7% of the total mutations seen in DMD, respectively [4,5]. The remaining mutations involve single nucleotide variants, small deletions or insertions, single-base changes, and splice site changes in 25–35% of DMD cases and in 10–20% of BMD cases [15,23,24,25,26,27] (Table 1). Among them, nonsense mutations are observed in 20–30% of DMD cases and ~5% of BMD cases [15,17].

If MLPA detects a single exon deletion, it is likely that a point mutation or a small deletion exists within the deleted exon; therefore, direct sequencing methods are needed to confirm diagnosis [28,29] (Table 1). If a mutation is not detected, Western blot analysis and/or immunohistochemistry would be required to identify and quantify the protein from muscle biopsies [28,29] (Table 2). If protein studies show a decrease in dystrophin expression, direct sequencing of all exons and nearby introns is warranted. Chromosomal microarray analysis is considered in rare cases with a contiguous gene deletion [30].

So far, therapy aimed at complete recovery has not been developed for these diseases; however, advances in molecular biology have enabled the development of various therapeutic strategies. In particular, exon skipping therapy to recover the reading frame [31], readthrough therapy for nonsense mutations [32], gene therapy using viral vectors [33], and cell transplantation therapy are being actively developed [34]. Here, I describe the advances in molecular therapy for DMD based on *DMD* gene mutations.

Sometimes mutational information is insufficient to distinguish between DMD and BMD. In such cases, protein analysis by western blotting and/or immunohistochemistry should be performed. Identification of mutations is essential, not only in determining the most effective therapy and disease management for the patient, but also for familial counseling to establish carrier status, which is strongly suggested [29]. Accurate and early diagnosis allows for evaluation of cardiorespiratory function and the cardiopulmonary state, professional management of physical therapy, and application of new therapies [29].

## 3. Pathomechanism of Dystrophinopathy

Dystrophin assumes a rod-like structure comprised of an N-terminal actin-binding domain; a rod domain with repeating arrangements of 24 spectrin-like motifs; a cysteine-rich domain, which binds dystroglycan/sarcoglycans complexes; and a C-terminal domain, which interacts with the syntrophin complex dystrobrevin [35,36]. Dystrophin binds to dystroglycans, sarcoglycans, and syntrophin complexes, forming a dystrophin glycoprotein complex (DGC). DGC binds the cytoskeleton and extracellular matrix in skeletal muscle cells [35,36] and plays a role not only as a membrane stabilizer, but also as a signal converter from the extracellular environment to the cytoplasm via receptors on muscle cells prior to contraction [2]. Dystrophin deficiency caused by *DMD* gene mutations lack sarcolemmal stability against mechanical stress [37,38]. As a result, an increase in intracellular calcium ion concentration activates calcium-dependent proteases, such as calpain and various chemokines/cytokines, resulting in muscle degeneration and necrosis, eventually leading to fibrosis and fatty cell infiltration [39,40] (Figure 1). In addition, the abnormal expression and/or activity of various molecules such as neuronal nitric oxide synthase (nNOS) [41,42], aquaporin 4 [43,44], sodium ion channels [45], L-type calcium ion channels [46], or the stretch-activated calcium channel (located in the sarcolemma) [47,48] may be related to muscle degeneration. Furthermore, nitric oxide (NO) may also impact the regulation of calcium signaling in a muscle-protective way and immune responses [49]. Several studies have reported that modulation of NO signaling may alleviate dystrophin-deficient muscle function and pathology [50,51,52,53,54].

## 4. DMD Animal Models for the Development of Mutation-Specific Therapeutic Strategies

DMD requires treatment given its severe symptoms and clinical outcomes. To elucidate pathogenesis and develop novel therapies, it is necessary to utilize appropriate animal models exhibiting DMD symptoms and pathology. For DMD research, *mdx* mice, *mdx*52 mice, and dystrophic dogs have mainly been used [55].

### 4.1. Mdx Mice

*Mdx* mice with a C57BL/10ScSnJ background harbor a nonsense mutation in exon 23 of the *Dmd* gene [56]. Compared with DMD patients, limb muscle weakness is mild; however, respiratory muscles, such as the diaphragm, prominently indicate necrosis, degeneration, and fibrosis [57]. In general, *mdx* mice develop heart failure from approximately 6 months of age onwards [58], and aged *mdx* mice show the myocardial fibrosis and dysfunction observed in DMD patients [59]. *Mdx* mice have been bred in many laboratories around the world for elucidation of DMD pathogenesis and development of therapeutic strategies.

### 4.2. Mdx52 Mice

*Mdx*52 mice that lack exon 52 of the *Dmd* gene were developed using gene knockout technology [60]. These mice lack dystrophin and show dystrophic changes along with muscle hypertrophy. Unlike *mdx* mice, *mdx*52 show an abnormal electroretinogram resulting from a deficiency of the dystrophin isoform Dp260, which is expressed in the retina [61]. To date, *mdx*52 mice have been used for the development of exon 51, exon 53, and multiexon 45–55 skipping strategies that will be described in Section 5.

### 4.3. Dystrophic Dogs

Golden retriever muscular dystrophy (GRMD), found in the Golden Retriever breed of dog, shows progressive muscular wasting, electrocardiography abnormalities, and myocardial degeneration, similar to DMD in humans [62,63]. The National Center of Neurology and Psychiatry in Japan has established a colony of Beagle-based dystrophic dogs (canine X-linked muscular dystrophy in Japan (CXMD_J_)) inseminated by GRMD sperm, for easy breeding and testing [64]. These dystrophic dogs harbor a point mutation in the intron 6 splice acceptor site causing exon 7 skipping, resulting in a stop codon in exon 8 and thereby inhibiting dystrophin production [65].

Unlike DMD patients and *mdx* mice, serum CK levels after birth are extremely high in dystrophic dogs [63,66]. Approximately 30% of dystrophic dogs die neonatally due to respiratory failure or feeding difficulty [63,66]. At 2–3 months of age, the dogs show muscle wasting of limbs and temporal muscles, limb–joint contractures, gait disturbance, macroglossia, and dysphagia [63,66]. Cardiac involvement is detected as abnormal Q waves on electrocardiography (ECG) around two months of age, and fibrosis of the left ventricle is detected by cardiac ultrasonography and pathological examination after 12 months of age [67]. Compared with mouse models, the phenotypic severity in dystrophic dogs is more similar to DMD. Thus, dystrophic dogs are useful for the elucidation of pathogenesis and development of therapy.

Cavalier King Charles Spaniels with dystrophin-deficient muscular dystrophy (CKCS-MD) have also been identified [68]. CKCS-MD dogs have a 5′ splice site missense mutation in intron 50 of the *DMD* gene, causing out-of-frame skipping of exon 50 and resulting in a lack of dystrophin and a severe dystrophic phenotype resembling DMD; therefore, this dog model is available to develop exon skipping therapy for DMD with hotspot mutations.

## 5. Therapeutic Strategies Based on Mutations of DMD

In the last two decades, advances in the management of the cardiopulmonary function of DMD patients have extended life expectancy by more than 10 years of what it was previously. Moreover, corticosteroids are administered as standard treatment, prolonging normal motor function and allowing for rehabilitation. Nevertheless, corticosteroids cannot stop the progression of DMD, and their long-term use causes debilitating side effects, which necessitates alternative therapeutic strategies. Based on the dystrophic mechanism, the most appropriate therapy may be to complement or restore dystrophin by targeting all organs that express it. Strategies for dystrophin protein restoration include (1) vector-mediated mini- or microdystrophin gene delivery, (2) nonsense readthrough therapy, and (3) exon skipping with synthetic antisense oligonucleotides (AOs) or genome editing.

### 5.1. Gene Therapy Using Vectors

Gene replacement therapy is considered a potential strategy for the treatment of DMD, aiming to restore the missing protein. Gene therapy for DMD involves expressing dystrophin by introducing it to muscle cells using a full-length or functionally truncated *DMD* gene (cDNA) in a viral vector. Representative vectors commonly used in nondividing muscle cells include lentiviral, adenoviral, adeno-associated virus (AAV), and human artificial chromosome (HAC) vectors. All are capable of long-term expression of the exogenous gene without eliciting overt host immune reactions. Viral gene transfer of the full-length *DMD* gene may restore wild type functionality; however, this approach is restricted by the limited capacity of recombinant viral vectors. The loading capacity of lentiviral, adenoviral, and AAV vectors is ~8 kb, 8–36 kb, and ~4.7 kb, respectively [69]. The large size of *DMD* gene transcript has been a major obstacle in developing methods for DMD gene therapy. However, numerous studies in both animal models and the clinic have generated considerable knowledge regarding the structural domains of dystrophin and have enabled rational design of highly functional mini- and microdystrophins more amenable to gene therapy applications. To impart the function of full-length dystrophin, mini- and microdystrophins retain the N-terminal actin-binding, cysteine-rich, and C-terminal domains, whereas the rod domain is truncated (Figure 2).

#### 5.1.1. Lentiviral Vector 

Lentiviral vectors, derived from HIV-1, can integrate into the host genome and achieve long-term transgene expression in a wide variety of dividing and nondividing cells, including skeletal muscle [70,71]. VSV-G-pseudo-typed lentiviral vectors can transduce adult mouse skeletal muscle cells—albeit with a relatively low efficiency [72]. Nonetheless, myofibers transduced with a fully functional 6.3-kb Becker-like dystrophin cDNA (minidystrophin gene) were partially protected from degeneration for at least six months in *mdx* mice [70,72]. 

Viral gene transfer of full-length dystrophin may restore wild type functionality. Lentiviral vectors can package and deliver inserts of a similar size to dystrophin. A recent breakthrough demonstrated that lentiviruses can be used to deliver full-length dystrophin to DMD myoblasts as a proof-of-concept, ex vivo gene therapy strategy [73,74]. However, lentiviral vectors come with biosafety issues, such as insertional mutagenesis by enhancer-mediated dysregulation of neighboring genes or aberrant splicing, immunogenicity of vector particles, toxicity of the transgene, and potential vertical or horizontal transmission by replication competent retroviruses [75].

#### 5.1.2. Adenoviral Vector

The adenoviral vector, a recombinant adenovirus, has been used to deliver a minidystrophin gene via injection into dystrophic muscles [76,77]. Minidystrophin confers important functional and structural protection of the limb muscles against high mechanical stress, even after partial somatic gene transfer [78]. Moreover, the expression of recombinant human minidystrophin cDNA was found to significantly reduce the myopathic phenotype in transgenic *mdx* mice [78]. Adenoviral vectors can efficiently transduce a broad variety of different cell types and have been used extensively in preclinical and clinical studies. However, adenoviral vectors retain residual adenoviral genes that contribute to inflammatory immune responses and toxicity. Moreover, these vectors often result in transient expression of the potentially therapeutic transgene. The latest generation of high-capacity adenoviral vectors is devoid of viral genes, has a significantly improved safety profile, and yields prolonged transgene expression compared with that of early generation vectors [79]. Nevertheless, transgene expression gradually declines over time even when high-capacity adenoviral vectors are used, possibly due to the gradual loss of vector genomes. Despite their improved safety, high-capacity adenoviral vectors can still trigger transient toxic effects in animals and patients [79]. Currently, integrating adenoviral vectors carry a malignancy risk due to their ability to integrate randomly into the target genome.

#### 5.1.3. AAV Vector

AAV vectors have the highest safety and transfection efficiency in skeletal muscle cells [33]. However, the AAV vector can only incorporate a maximum length of 4.9 kb. Thus, the shorter microdystrophin (CS1), embedded in the rod domain, is produced and incorporated in AAV vectors [80]. After binding to the receptor of the muscle membrane, CS1—which was introduced into the AAV vector—is incorporated into the cells and migrates to the nucleus. The gene is then translated in the nucleus and microdystrophin is expressed and localized to the cell membrane. When CS1 incorporated into type 2 AAV (AAV2) was administered to *mdx* mice, muscle pathology and muscle tension improved [81]. However, AAV2 carrying CS1 did not lead to dystrophin production in the skeletal muscle of dystrophic dogs due to strong immune responses [82]. CS1 incorporated into type 8 AAV vector (AAV8), whose T cell activation is lower than in AAV2, showed a high-efficiency expression in dystrophic dogs for eight weeks [83]. CS1 combined with type 9 AAV vector (AAV9) was also successfully introduced into cardiac [84,85] and whole-body skeletal muscles of neonatal dystrophic dogs [86].

In other AAV-based therapies, AAV-mediated gene therapy with *GALGT2* is currently being investigated to upregulate the expression of dystrophin and laminin-α2 surrogates, including utrophin, plectin1, agrin, and laminin α5 [87,88,89]. Phase I/IIa gene transfer clinical trials of this strategy for DMD are currently ongoing [89]. 

#### 5.1.4. HAC Vector

HACs have the capacity to deliver extremely large genetic regions to host cells without integration into the host genome, preventing insertional mutagenesis or genomic instability. A HAC vector with the entire human *DMD* gene (DYS-HAC) can be stably maintained in mice and in human immortalized mesenchymal stem cells (hiMSCs). In chimeric mice generated from embryonic stem cells transferred from DYS-HAC, isoforms of the DYS-HAC-derived human dystrophin were correctly expressed in a tissue-specific manner [90].

Furthermore, to the extent that the proliferative potential can withstand clonal DMD satellite cell-derived myoblast and perivascular cell-derived mesoangioblast expansion after HAC transfer, one study reported reversible cell immortalization mediated by lentiviral vectors via excisable hTERT and Bmi1 transgene delivery that extended cell proliferation. The cells remained myogenic in vitro and engrafted into murine skeletal muscle upon transplantation. These results represent next-generation HACs, capable of delivering reversible immortalization, complete genetic correction, additional dystrophin expression, inducible differentiation, and controllable cell death [91]. This HAC method holds promise for future DMD gene therapies.

### 5.2. Readthrough Therapy

Readthrough therapy restores dystrophin expression through strategies that inhibit translation termination at nonsense mutations. For example, the antibiotic gentamicin can skip (i.e., readthrough) a nonsense mutation in vitro [92]. Thus, gentamycin was injected into *mdx* mice with a nonsense mutation, which restored dystrophin expression [93]. This therapy is applicable to approximately 10% of DMD patients. However, gentamicin can cause nephrotoxicity and auditory nerve toxicity, and thus large and long-term administration is unadvisable for humans. Therefore, a new readthrough compound—PTC124—was developed and its administration to *mdx* mice restored dystrophin expression in skeletal and cardiac muscles [32]. Furthermore, PTC124 (Ataluren, Translarna^TM^) showed efficacy and safety during a Phase IIa trial [94,95]. Thus, Ataluren—produced by PTC Therapeutics Inc., South Plainfield, NJ, USA—has received conditional approval in the European Union for the treatment DMD patients with nonsense mutations [96]. The antibiotic arbekacin sulfate (NPC-14) also exhibits a similar mechanism to PTC124, and Phase II clinical trials are currently underway in Japan [97].

### 5.3. Exon Skipping Therapy

In DMD patients and animal models, some dystrophin-positive fibers (revertant fibers) are observed at the sarcolemma, despite the fact that it usually lacks dystrophin. The number of revertant fibers increases with age due to the degeneration and regeneration cycles of the disease [98,99]. The basic idea of exon skipping therapy is the skipping of exon(s) around the original mutation, which leads to correction of the reading frame and restoration of dystrophin expression at the sarcolemma. Thus, exon skipping strategies have attracted attention as potential DMD therapy [99,100,101].

The development of several new AOs has contributed to the advancement of DMD exon skipping therapy [102]. Moreover, the ethical issues involved in exon skipping therapy are fewer than those associated with vector-mediated gene therapy or stem-cell transplantation therapy. This is because AOs are classified as drugs, rather than gene therapy agents, by the Food and Drug Administration (FDA) of the USA and representative agencies in the European Union and Japan. Exon skipping therapy may be useful for up to 90% of DMD patients with deletion mutations [103,104]. Additionally, patients exhibiting very mild phenotypes with high blood CK concentrations have an in-frame deletion in the *DMD* gene [105,106]; therefore, it may also be possible to aim to make a mild phenotype more intense after exon skipping therapy.

#### 5.3.1. Exon Skipping Using Antisense Oligonucleotides

Antisense oligonucleotides are chemically synthesized, short (~20 bases) nucleic acids that are designed to hybridize to complementary pre-mRNA sequences [105]. Antisense oligonucleotides bind to an exon–intron boundary or exonic splicing enhancers (ESEs) to regulate a splice-suppressing spliceosome composed of proteins, snRNA, and mRNA [107,108]. The representative AO compounds 2′-*O*-methyl-phosphorothioate AO (2′OMeAO) and phosphorodiamidate morpholino oligomers (PMO or morpholino) are most frequently used because of their safety and efficiency [109,110]. Moreover, peptide-conjugated PMO (PPMO) and vivo-morpholino were developed to provide high uptake efficiency by cells [108,109]. The ESE finder program can be used to predict/search for ESEs [107] and pre-mRNA secondary structures during AO design [110].

The therapeutic strategy for DMD involves skipping an exon with a mutation or nearby exons via AOs during mRNA splicing, leading to the correction of the reading frame and restoration of dystrophin expression (Figure 3) [107]. It has been shown that exon skipping therapy converts the DMD phenotype to the BMD phenotype. However, the scenario is complex due to the phenotypic diversity of BMD, ranging from asymptomatic to symptoms typical of severe DMD; furthermore, cardiac involvement of BMD frequently progresses faster than DMD. Therefore, I have suggested that researchers aim at improving dystrophinopathy to a very mild or asymptomatic phenotype prior to administration of this therapy [108]. For a successful therapeutic strategy, it is important to understand the BMD phenotypes of in-frame mutations and dystrophin structure after exon skipping, rather than simply correcting the reading frame by AOs [108].

#### 5.3.2. Preclinical Trials of Exon skipping Therapy using Antisense Oligonucleotides

Preclinical trials have been performed using DMD model mice (*mdx* and *mdx*52) and dogs. In *mdx* mice, 2’OMeAO targeting of the exon–intron boundary sequence of exon 23 of the *DMD* gene successfully induced more than 20% dystrophin-positive skeletal muscle fibers [111]. Moreover, systemic administration of PMO targeting the same region revealed approximately 25% dystrophin-positive skeletal muscle fibers [106,112]. Long-term administration of PMO yielded more than 70% dystrophin-immunoreactive fibers in the quadriceps femoris and gastrocnemius muscles and improved muscle pathology and muscle tension [106,111]. A cocktail of three kinds of PMOs targeting exons 6 and 8 restored dystrophin expression in whole-body skeletal muscles after being administered to dystrophic dogs; the results showed improved muscle pathology, magnetic resonance imaging findings, and motor function as well as decreased serum CK levels [112]. However, in other mouse and dog studies, restoration of dystrophin expression in cardiac muscle was lower than in skeletal muscles. Hence, PPMO, which exhibits high cell penetration efficiency, was used to increase the restoration rate of dystrophin [113,114].

Exon skipping therapy is a personalized medicine based on gene mutations; targeting the same exon for a greater number of patients can reduce production costs and medical expenses. Deletion mutations in the *DMD* gene are concentrated in exons 3–7 and exons 45–55 [4,5] The number of DMD patients with a deletion in exons 45–55 account for ~60% of the DMD population [115]. Approximately 13% of DMD patients can be treated when exon 51 is targeted by skipping [116]. Thus, *mdx*52 mice were used for the development of an exon 51-skipping strategy in preclinical trials; as a result, the systemic administration of PMO recovered dystrophin expression in skeletal muscle and improved muscle pathology and function of mice [117]. In addition, exon 53 is the second most commonly targeted exon in patients, whose skipping strategy has been previously reported [118].

#### 5.3.3. Clinical Trials of Exon Skipping Therapy using Antisense Oligonucleotides

For patients with DMD, clinical applications of exon 51 skipping were first performed as a focal injection of 2’OMeAO. Results from intramuscular injections of PRO051 into the tibialis anterior (TA) muscle revealed the presence of sarcolemmal dystrophin in more than 50% of muscle fibers. The amount of dystrophin in total protein extracts ranged from 3 to 12% of controls, and a quantitative ratio of dystrophin to laminin-α2 was 17 to 35% of control specimens [119]. During the systemic administration of PRO051 (2’OMeAO) to DMD patients (phase I/IIa), weekly abdominal subcutaneous injections were administered for 12 weeks in 2.0 to 6.0 mg/kg doses. The results showed that dystrophin was expressed in approximately 60–100% of muscle fibers in 10 of the 12 trial participants. Furthermore, dystrophin expression increased in a dose-dependent manner to 15.6% of normal controls. A six-minute walk test (6MWT) showed a mean improvement of 35.2 ± 28.7 m [120].

A 48-week double-blind, placebo-controlled, multicenter trial of PRO051 (drisapersen) subcutaneous administration (6 mg/kg) was performed with 53 patients (18 received once-weekly injections, 17 received intermittent drisapersen (nine doses over 10 weeks), and 18 received a placebo (either continuous or intermittent)). Mean distance of the 6MWT at week 25 increased by 31.5 m for patients receiving continuous drisapersen, with a mean difference of 35.09 m compared with that of patients given the placebo. As the studies extended beyond the 25th week, the difference between treated and placebo cohorts were less apparent and no longer statistically significant (*p* = 0.051) by week 49 [121]. Drisapersen, developed by BioMarin Ltd., was discontinued at the end of a phase III study due to inefficacy and adverse effects, such as injection site reactions, mild proteinuria, and rarely reduced platelet counts. The product was further tested in a GSK/Prosensa sponsored, pivotal, phase III, placebo-controlled clinical trial that included 186 DMD patients. Boys with DMD were randomized into groups and given a dose of drisapersen at 6 mg/kg/week (*n* =125) or a placebo (*n* = 61) for 48 weeks. However, the study failed to achieve statistical significance in its primary end point, the 6MWT [122]. Thus, the US FDA voted that there was no conclusive benefit, and BioMarin Ltd. withdrew the compound from clinical testing. Presently, there are no clinical trials that employ the use of drisapersen or the related 2OMeAO.

As for the use of PMO in exon 51 skipping, intravenously delivered AVI-4658 (eteplirsen) was performed on 19 ambulatory boys (5–15 years old) with DMD in an open-label, phase II, dose-escalation study [123]. Twelve weekly and intravenously administered doses of eteplirsen induced exon 51 skipping, leading to dystrophin expression that increased in a dose-dependent manner. Seven patients responded to treatment, demonstrating significantly increased dystrophin fluorescence intensity [124,125]. Another clinical trial investigating eteplirsen, under the sponsorship of Sarepta Therapeutics Ltd., was designed as a phase IIb, double-blind, placebo-controlled study [126]. Twelve boys with DMD (7–13 years old) were enrolled and divided into three cohorts (placebo, 30 mg/kg/week, and 50 mg/kg/week). This study is currently ongoing as an open-label, long-term study, inclusive of safety and efficacy outcomes with evaluation of long-term dystrophin expression at 180 weeks and functional motor assessments, including the 6MWT (36 months). The first phase of eteplirsen administration revealed an increase in dystrophin expression, but the findings were considered inconclusive [127]. Muscle biopsies were performed at 180 weeks for 11 patients and three methods used to quantify total dystrophin protein (Western blot analysis, scoring of dystrophin stained sections by three independent pathologists, and analysis of immunofluorescent dystrophin fiber intensity). The results showed significant increases in dystrophin expression in eteplirsen-treated samples, with Western blot analysis showing an absolute increase in expression at + 0.85% difference from that of control, representing an 11.6-fold increase (*p* = 0.007). Motor assessment resulting from long-term use of eteplirsen was evaluated by the 6MWT, pulmonary functions of maximum inspiratory pressure, maximum expiratory pressure, and forced vital capacity. Safety assessments of adverse events in the eteplirsen study were also recorded [128,129]. Eteplirsen improved the 6MWT distance by 151 m compared with that of historical age and mutation-matched controls for three years of follow-up. Furthermore, weekly eteplirsen infusions were well tolerated, with no serious adverse events leading to treatment interruption or dose change [128]. Thus, eteplirsen has received accelerated approval from the FDA; however, the approval is preliminary and requires further clinical confirmation of efficacy in the currently ongoing clinical trials.

The National Center of Neurology and Psychiatry in cooperation with Nippon Shinyaku Co, Ltd. (Tokyo, Japan) developed a PMO (NS-065/NCNP-01) for exon 53 skipping [130]. This therapy is applicable to 10.1% of patients with DMD. The phase I study was an open-label, dose-escalation clinical trial for determining the safety, pharmacokinetics, and activity of this PMO. NS-065/NCNP-01 at doses of 1.25, 5, or 20 mg/kg were administered weekly for 12 weeks to 10 DMD patients (6–16 years old). Severe adverse effects were not observed and muscle biopsies revealed that NS-065/NCNP-01 induced exon 53 skipping in a dose-dependent manner and increased dystrophin expression in seven of 10 patients. Thus, NS-065/NCNP-01 appears safe, has pharmacokinetic potential, and is currently being tested in a phase II clinical trial [130]. Other ongoing clinical trials of exon skipping therapy include SRP-4053 for exon 53 targeting (Golodirsen; Sarepta Therapeutics Ltd., Cambridge, MA, USA), exon 45 targeting by SRP-4045 (Casimersen; Sarepta Therapeutics Ltd.), and exon 45 or DS-514b targeting using 2′-*O*,4′-*C*-ethylene-bridged nucleic acid (ENA; Daiichi Sanko Co., Ltd., Chuo-ku, Tokyo, Japan) among others.

#### 5.3.4. Issues and Prospects of Exon Skipping Therapy using AOs

There are some issues associated with the development of exon skipping therapy. First, from the study of animal models, the effect of exon skipping lasts 2–3 months; making repeated administration necessary [109]. Second, skipping efficiency varies among different tissues and cells; for example, less dystrophin is induced in the cardiac muscle. Studies on mouse-derived C2C12 myoblasts and laminin-α2 chain-null congenital muscular dystrophy mice revealed that PMO is easily incorporated during active muscle regeneration [131]. Moreover, PPMO can be incorporated into cells via scavenger receptor class A-mediated endocytosis after forming spontaneous micellar nanoparticles [132]. Thus, strategies should aim to improve AO delivery. Third, exon skipping therapy only targets one exon; therefore, the number of patients that can be treated is limited. Furthermore, very little is known about the converted phenotype following exon skipping therapy due to our lack of knowledge regarding dystrophinopathy patients with in-frame mutations. Nevertheless, it has been reported that patients with a deletion of exons 45–55 show a very mild or asymptomatic phenotypes [115,133,134]. Exons 45–55 span an entire region of genetic hotspots; thus, exon skipping therapy of exons 45–55 for patients with DMD or severe BMD with mutations in this region would result in approximately 60% of DMD patients that can be converted to very mild or asymptomatic phenotypes, making this a highly effective therapeutic strategy [109,115]. This strategy was successful in *mdx*52 mice, showing restoration of dystrophin expression and improved muscle pathology and function after multiexon skipping treatment with PMOs [135]. Furthermore, given that a case with a deletion of exons 3–9—spanning a second genetic hotspot region (exons 3–7)—was asymptomatic, we now understand that exon 3–9-skipping therapy is applicable for approximately 7% of DMD patients and may also convert them to a very mild phenotype [136]. However, the challenges of multiexon skipping therapy include attenuation of skipping efficiency due to AO double chain formation or off-target effects.

#### 5.3.5. Exon Skipping via Genome Editing

As mentioned above, AOs exhibit some drawbacks, such as poor uptake in the heart and short-lasting efficacy. To overcome these issues, clustered regularly interspaced short palindromic repeats (CRISPR)-associated protein 9 (CRISPR/Cas9)-mediated gene editing has recently become available for exon skipping therapy [137]. The first in vivo gene editing using CRISPR/Cas9 was performed to correct a nonsense mutation in exon 23 of the *Dmd* gene in *mdx* mice [138]. Components of CRISPR/Cas9 were injected into *mdx* zygotes at the single-cell stage, followed by zygote implantation into pseudopregnant mice. The obtained treated pups varied in the extent of *Dmd* gene editing. Approximately 50% of muscle fibers were dystrophin-positive—confirmed by immunostaining—in *mdx* mice at 7–9 weeks of age. Treated mice also showed decreased serum CK levels and improved grip strength.

AAV vectors (serotype 8 or 9) have been used to deliver CRISPR/Cas9 components in vivo, either intramuscularly, intraperitoneally, or intravenously to delete the mutated exon 23 of the *DMD* gene from the genome of *mdx* mice [139,140,141]. These studies showed restored dystrophin in both skeletal and cardiac muscles—confirmed by immunostaining and Western blotting—and was accompanied by relocalization of DGC components at the sarcolemma and improvements in skeletal muscle function. Off-target effects were not observed in these treatments. Moreover, CRISPR/Cas9-mediated multiexon skipping of exons 21–23 [142], 52–53 [143], and 45–55 [144] has been successful in DMD mouse models. These strategies can increase the number of patients for future therapeutic applicability. 

The CRISPR/Cas9 strategy was also applied to CKCS-MD dogs [145]. Cas9-containing AAV9 vectors were intramuscularly delivered to the TA muscles of 1-month-old CKCS-MD dogs (*n* = 2) and then treatment efficacy was evaluated six weeks postinjection. Dystrophin levels of approximately 52 or 67% of that of wild type were observed in the TA muscles of treated dogs, as evidenced by Western blot analysis; contralateral muscles injected with saline showed on average 2% dystrophin of wild type levels. Improvements in histopathology and restoration of β-dystroglycan expression were also observed in the treated muscles, and there was no evidence of immune response. Deep sequencing revealed no significant gene editing on the most likely off-target sites as a result of treatment. Moreover, eight weeks after the intravenous injection of systemically treated CKCS-MD dogs (1-month old; *n* = 2), dystrophin was found restored in various skeletal muscles as well as the heart, which was confirmed by immunostaining and Western blotting. Histopathological improvements and restored expression of β-dystroglycan were also observed, although muscle function tests were not conducted.

Human induced pluripotent stem cells derived from DMD have also been served as a development of therapy by CRISPR/Cas9 [146,147]. CRISPR/Cas9 technology essentially yields the same results as exon skipping with AOs, but with the advantage of inducing permanent corrections of the mutated *DMD* gene without requiring repetitive treatment—unlike with AOs. Nevertheless, it is necessary to establish therapeutic safety and efficacy in preclinical studies before embarking on clinical trials for DMD patients.

## 6. Future Aspects of Therapeutic Strategies

Despite being a single gene disorder, various therapeutic approaches have been developed against the background of the complex pathological mechanism of DMD. The most fundamental treatment for DMD is considered supplementation or recovery of dystrophin. In this regard, the use of viral vectors, exon skipping, or readthrough therapies as described in this review are desirable. However, each treatment approach also faces various difficulties. Although the use of viral vectors can be widely adapted regardless of the gene mutation, there is a limit to the size of genes that can be incorporated while also ensuring safety. Meanwhile, exon skipping therapy is tailor-made based on the genetic mutation; however, this limits the number of target patients, and there are other problems regarding introduction efficiency and symptom improvement. For readthrough therapy, the number of subjects that can be treated is also very limited. Moreover, in any treatment, the type and number of target organs (skeletal muscle, myocardium and smooth muscle, and nervous system) of DMD patients are both varied and high, which is another hurdle for these therapeutic strategies to overcome. Although these issues may be resolved in the future, it is necessary to combine several drug therapies.

## Figures and Tables

**Figure 1 jpm-09-00016-f001:**
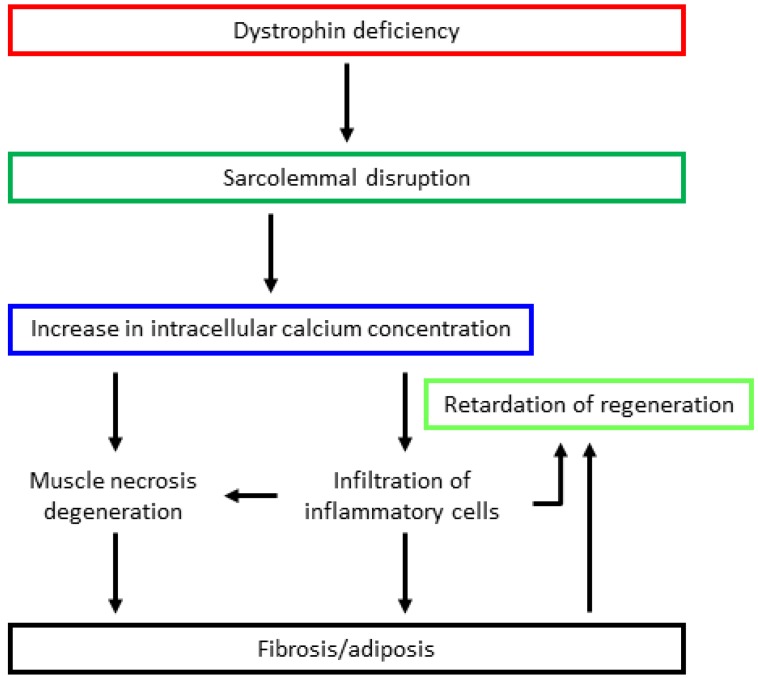
Hypothetical pathogenic mechanism of dystrophin deficiency. Dystrophin deficiency causes sarcolemmal disruption and calcium channel activation by mechanical stress. As a result, an increase in intracellular calcium through the concentration gradient and stretch-activated calcium channel activates calcium-dependent protease and chemokines/cytokines, resulting in muscle degeneration and necrosis. Following necrosis, muscle regeneration is retarded, which causes disease progression as regeneration cannot catch up with necrosis. Finally, fibrous and fatty tissues infiltrate the muscle fibers.

**Figure 2 jpm-09-00016-f002:**
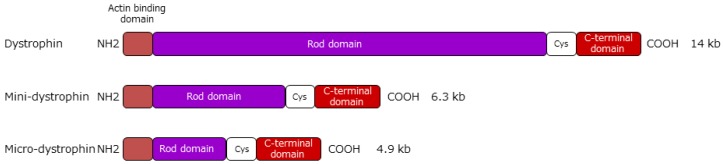
Structure of mini- and microdystrophins and utrophin. Minidystrophin is a dystrophin of mild BMD patients that is ~6.4 kb in size. Based on this, the rod region was further truncated to create microdystrophin (4.9 kb) that can be incorporated into viral vectors. Both dystrophins retain the actin-binding, cysteine-rich (Cys), and C-terminal domains of normal, full-length dystrophin (14 kb).

**Figure 3 jpm-09-00016-f003:**
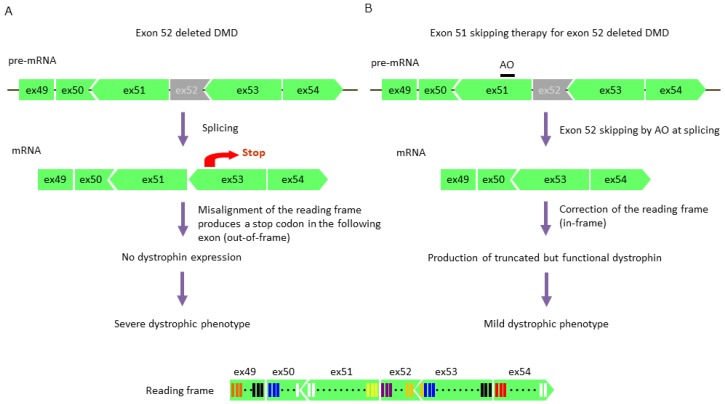
Exon 51-skipping therapy for exon 52-deleted Duchenne muscular dystrophy (DMD). (**A**) DMD with an exon 52 deletion. Exon 52 (the number of bases is not a multiple of 3) is absent from pre-mRNA, which results in a misalignment of the reading frame after splicing (out-of-frame); this leads to a premature stop codon in the following exon of the mRNA. As a result, dystrophin is not produced, leading to a severe dystrophic phenotype. (**B**) Exon 51-skipping therapy for exon 52-deleted DMD. Exon 51 is skipped by AO (sum of deleted bases is a multiple of 3) in pre-mRNA; subsequently, the reading frame is corrected in mRNA (in-frame) and translation progresses normally. As a result, a truncated but functional dystrophin is produced, resulting in a mild dystrophic phenotype. AO, antisense oligonucleotide; ex, exon.

**Table 1 jpm-09-00016-t001:** Diagnosis by DNA analysis of dystrophinopathy [28,29].

DNA Analysis	Mutational Type (Frequency)		Note
MLPA	Deletions of one or more exons (60–70%) Duplications (5–10%)		Mutational type: DMD: out-of-frame mutation BMD: in-frame mutation
Direct sequencing	Single nucleotide variants Small deletions or insertions Single-base changes Splice site changes	(20–35% in total)	Frequency: DMD: 25–35% BMD: 10–20%

MLPA: multiplex ligation-dependent probe amplification.

**Table 2 jpm-09-00016-t002:** Diagnosis by protein analysis of dystrophinopathy [28,29].

	Immunohistochemistry	Western Blotting	
		Molecular Weight (normal 427 kDa)	Protein Amount (normal 100%)
DMD	Compete or almost complete defect	Not detected	0–5%
BMD	Reduced staining and/or patchy pattern	Normal	20–50%
		Abnormal	20–100%

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
