# Peer review of "Mutation-Based Therapeutic Strategies for Duchenne Muscular Dystrophy: From Genetic Diagnosis to Therapy"

_jpm, 2019, doi:10.3390/jpm9010016_

Reviewer 1 Report

This is an overall good review article well  balanced as far as the gene therapy and molecular diagnosis are concerned. The part on the pharmacological approaches instead needs to be amended by the insertion of a series of important targets the authors have overlooked

Nitric oxide and NSAIDs-mitochondrial activity-deacetylase inhibitors-autophagy. Some of these (NO) may also impact on the regulation of calcium signalling in a muscle-protective way and immune responses

Author Response

1.      This is an overall good review article well balanced as far as the gene therapy and molecular diagnosis are concerned. The part on the pharmacological approaches instead needs to be amended by the insertion of a series of important targets the authors have overlooked. Nitric oxide and NSAIDs-mitochondrial activity-deacetylase inhibitors-autophagy. Some of these (NO) may also impact on the regulation of calcium signaling in a muscle-protective way and immune responses.

Response:

Thank the reviewer for the comment and suggestion. According the suggestion, I have referred the role of NO and cited the references in the text.

Reviewer 2 Report

This review entitled ‘Mutation-based Therapeutic Strategy for Duchenne Muscular Dystrophy: from genetic diagnosis to therapy’ covers mutations and diagnosis of dystrophinopathies, 3 animal models of DMD and therapeutic strategies; specifically, mutation-independent, viral vectors and read-through molecules. Whilst this review gives an in-depth insight into these topics, I have some concerns with the structure, use of referencing and novelty of the manuscript, as detailed below.

Referencing: My major concern with this manuscript is the haphazard use of referencing. Whilst the section on therapeutic strategies is fairly well referenced, sections 1-3 only contain 5 references – including when numbers are quoted. For example, in the 1st paragraph of the introduction the author does not reference any of the publications which describe clinical features, prevalence and incidence, problems with genetic counselling etc. There are also no references for information on dystrophin expression, associated proteins, pathological consequences and types of DMD mutations. This includes when specific statistics are used, an example of which can be seen in line 88. The tables also need referencing. This falls below publication standards and must be fully addressed for the publication to be considered.

Language: Several parts of this review are confusingly worded making it difficult to disseminate the underlying point the author is making. For example, ‘Further, DGC binds to the membrane through muscle plasma membranes from the cytoplasm of skeletal muscle cells’ And at lines 92-3 the increase in CK levels is not accidental. English grammar and phrasing needs to be checked throughout, which should clear up these points.

Repetition: there is a lot of repetition throughout the manuscript which if removed would make the manuscript easier to read and more concise. Care should be taken to avoid this. For example, sections 1-3 contains several mentions of the different locations and types of mutations that cause DMD. Additionally, in section 5.1 is ‘therapies based on the pathomechanism’, the first of which listed is viral vectors and exon skipping. Then in section 5.2 gene therapies and viral vectors are mentioned again. This whole section (5) could be made clearer if the subheadings were more defined. Clearer headings would also improve the clarity of section 5.1 as the development of therapies needs to be put into the context of current therapeutic options (i.e. corticosteroids) vs future potential therapies such as gene therapies and other therapeutic interventions only studied in pre-clinical animal models. Especially important as the focus of the manuscript is ‘mutation-based’ yet the majority of interventions here are mutation-independent.

Novelty: whilst well referenced, the therapeutic strategies section appears to be another review of the same strategies reviewed elsewhere. Additionally, several important therapeutic developments are entirely missing. For example, in 5.2 only micro-dystrophin is mentioned as a DMD viral vector strategy, and this needs to be distinguished from mini-dystrophin. Other AAV-based therapies are missing, such as use of utrophin modulation, such as AAV-mediated gene therapy with GALGT2, or put into context of current clinical trials. This also applies to 5.3 as only 1 of several clinical trials is mentioned. Also missing are preclinical models using DYS-HAC (Hoshiya et al Mol. Ther. 2009, Beneddetti et al EMBO 2018). Focussing on the newer forms of mutation-specific strategies and recent progress/ failures would improve the novelty of this manuscript. For example, the use of CRISPR briefly mentioned in the last paragraph.

Other specific points to be addressed:

-        The statement that serum CK levels can be used for diagnosis of DMD is not backed up by a reference and is also generally not an accepted method within the field, as CK levels vary between different ages, ethnicities and sexes. Instead, CK levels can be used to identify a muscle disorder, but the diagnosis of DMD must be performed with genetic testing. The author should amend this statement.

-        In figure 1 I do not agree with the line between calcium channel activation directly causing retardation of regeneration. Please provide references to support.

-        The models of DMD section only takes into account mdx mice and GRMD dogs. There are in fact several other models, including the utrophin-mdx mouse model which is a widely used murine model (Deconnick et al Cell 1997), with other models described in detail in McGreevy et al Dis Model Mech. 2015). However, as the review is about mutation specific strategies for DMD, this section only needs to be included if it is in the context of the title theme.

-        Clarity should be made between DMD and dystrophinopathies, which appears as a term in later sections. Please define earlier.

Author Response

1.      This review entitled ‘Mutation-based Therapeutic Strategy for Duchenne Muscular Dystrophy: from genetic diagnosis to therapy’ covers mutations and diagnosis of dystrophinopathies, 3 animal models of DMD and therapeutic strategies; specifically, mutation-independent, viral vectors and read-through molecules. Whilst this review gives an in-depth insight into these topics, I have some concerns with the structure, use of referencing and novelty of the manuscript, as detailed below.

Response:

Thanks the reviewer for the appropriate comments, critics, and suggestions. I have taken these into the consideration to revise the manuscript.

2.      Referencing: My major concern with this manuscript is the haphazard use of referencing. Whilst the section on therapeutic strategies is fairly well referenced, sections 1-3 only contain 5 references – including when numbers are quoted. For example, in the 1st paragraph of the introduction the author does not reference any of the publications which describe clinical features, prevalence and incidence, problems with genetic counselling etc. There are also no references for information on dystrophin expression, associated proteins, pathological consequences and types of DMD mutations. This includes when specific statistics are used, an example of which can be seen in line 88. The tables also need referencing. This falls below publication standards and must be fully addressed for the publication to be considered.

Response:

Thanks the reviewer for the comments. I have referenced the publications concerning clinical features, prevalence and incidence, problems with genetic counseling, information on dystrophin expression, associated proteins, pathological consequences, types of DMD mutations, and Table 1.

3.      Language: Several parts of this review are confusingly worded making it difficult to disseminate the underlying point the author is making. For example, ‘Further, DGC binds to the membrane through muscle plasma membranes from the cytoplasm of skeletal muscle cells’ And at lines 92-3 the increase in CK levels is not accidental. English grammar and phrasing needs to be checked throughout, which should clear up these points.

Response:

According to the reviewer’s comments, I rewrote the parts corresponding and checked throughout.

4.      Repetition: there is a lot of repetition throughout the manuscript which if removed would make the manuscript easier to read and more concise. Care should be taken to avoid this. For example, sections 1-3 contains several mentions of the different locations and types of mutations that cause DMD. Additionally, in section 5.1 is ‘therapies based on the pathomechanism’, the first of which listed is viral vectors and exon skipping. Then in section 5.2 gene therapies and viral vectors are mentioned again. This whole section (5) could be made clearer if the subheadings were more defined. Clearer headings would also improve the clarity of section 5.1 as the development of therapies needs to be put into the context of current therapeutic options (i.e. corticosteroids) vs future potential therapies such as gene therapies and other therapeutic interventions only studied in pre-clinical animal models. Especially important as the focus of the manuscript is ‘mutation-based’ yet the majority of interventions here are mutation-independent.

Response:

Thank the reviewer for the appropriate comments. I greatly changed the constitution of the paragraphs and sentences of the applicable points.

5.      Novelty: whilst well referenced, the therapeutic strategies section appears to be another review of the same strategies reviewed elsewhere. Additionally, several important therapeutic developments are entirely missing. For example, in 5.2 only micro-dystrophin is mentioned as a DMD viral vector strategy, and this needs to be distinguished from mini-dystrophin. Other AAV-based therapies are missing, such as use of utrophin modulation, such as AAV-mediated gene therapy with GALGT2, or put into context of current clinical trials. This also applies to 5.3 as only 1 of several clinical trials is mentioned. Also missing are preclinical models using DYS-HAC (Hoshiya et al Mol. Ther. 2009, Beneddetti et al EMBO 2018). Focussing on the newer forms of mutation-specific strategies and recent progress/ failures would improve the novelty of this manuscript. For example, the use of CRISPR briefly mentioned in the last paragraph.

Response:

Focusing on the newer forms of mutation-specific strategies and recent progress/ failures would improve the novelty of this manuscript. For example, I have added the sentence concerning the use of CRISPR briefly mentioned in the last paragraph.

Other specific points to be addressed:

1.      The statement that serum CK levels can be used for diagnosis of DMD is not backed up by a reference and is also generally not an accepted method within the field, as CK levels vary between different ages, ethnicities and sexes. Instead, CK levels can be used to identify a muscle disorder, but the diagnosis of DMD must be performed with genetic testing. The author should amend this statement.

Response:

2.      In figure 1 I do not agree with the line between calcium channel activation directly causing retardation of regeneration. Please provide references to support.

Response:

I made modifications because the applicable point of my figure was jumped to, as the reviewer pointed-out.

3.      The models of DMD section only takes into account mdx mice and GRMD dogs. There are in fact several other models, including the utrophin-mdx mouse model which is a widely used murine model (Deconnick et al Cell 1997), with other models described in detail in McGreevy et al Dis Model Mech. 2015). However, as the review is about mutation specific strategies for DMD, this section only needs to be included if it is in the context of the title theme.

Response:

The review is about mutation specific strategies for DMD; therefore, this section only needs to be included, and I have changed the title theme of this section.

4.      Clarity should be made between DMD and dystrophinopathies, which appears as a term in later sections. Please define earlier.

Response:

Thank the appropriate comment. I have defined the term of dystrophinopathy earlier.

Reviewer 3 Report

Overall comment:

The manuscript needs thorough editing of English language. Many sentences are incorrect due to incorrect use of English i.e. line 41 (use of noticed), line 47 patients have a de novo mutation.

Major points:

The review is not focused. Based on the title the review is about therapies which restore mutations in the DMD gene from diagnosis to therapy. A large part of the review deals however with ‘other non-related issues’ and clear structure is lacking. All information regarding therapies which target ‘secondary pathology’ is not relevant for this particular review. The author only refers to a few selected strategies, without really explaining. It does not add anything to the review. I would advice to delete it: Figure 1B and all text about this (section 5.1).

The review is about targeting the DMD gene to restore dystrophin expression. The author only hints at the CRISPR/Cas9 developments in line 307. Explaining this should deserve a separate paragraph.

The table 1 needs explanation.

Some parts of the text are very confusing for non-expert readers. An example: the entire DMD gene does not fit in a viral vector (but only 4.9 kb as mentioned in line 195). Author start the paragraph however with … by introducing it to muscle cells using a full-length or functionally truncated DMD gene (cDNA) into a viral vector (line 191-192). Only 3 sentences later, the reader will read that what was just said (inserting a full-length cDNA) is in fact not possible. Rewrite this part and be more clear from the start onwards. Author refers to both mini and micro dystrophin. This difference should be explained. The part about gene therapy using viral vectors also lacks the more recent developments. There is no reference at all made to the work done so far in DMD patients with micro- and mini-dystrophin. In addition, it would be good for a reader to understand the proc and conc of each of the described therapeutic strategies. For each of the therapeutic strategies pictures explaining the technique should be provided like is done for exon skipping. (i.e. for this part; which mini, micro-dystrophins have been used in which clinical trial etc.).

Also the order of the paragraphs would make more sense if it first deals with therapies which have conditional approval followed by those which are still in phase I/II.

The information provided about PTC124 is unclear for non-experts. There is no reference made in the text that PTC124 is the same as Ataluren and Translarna. Now sentence 220-221 make no link to PTC124 for readers who are not aware that this is the tradename. Also in sentence 218 it is mentioned that use was prevented due to initial trial outcomes. However, this was later changed. Please update the text.

Line 229: it is indeed expected that depending on the mutation that will be corrected and the size and make-up of the synthesized dystrophin protein, there will be differences in therapeutic outcome. However, it is not expected that DMD patient will become asymptomatic. This is misleading information and should be removed. Especially since the best dystrophin levels restored so far was 0.9% of wildtype. We are very far from treating a patient such that the phenotype will be asymptomatic. Author should remove all references to making a DMD patient asymptomatic.

Figure 2: the reading frame which is shown does not match with the reading frame shown in the examples, and it will not be clear to non-experts whether this is the situation of A or B. Please adapt so that both matches.

The focus of the paragraphs starting at line 252 is skewed. Thirteen lines of text are on the pre-clinical development, while only 4 lines are on the trials conducted in patients and the FDA filings. No reference on data obtained, patient numbers used, trial setup etc. is made.

Minor points:

Author consistently refers to muscle atrophy in DMD. In patients the calf show muscle hypertrophy. The rest of the skeletal muscles are wasted over time, but they are not atrophic.

Line 136; mdx mice do develop heart failure from ~6 months of age onwards. The reference the author use is studying young mice, and does not study heart function by advanced techniques like MRI.

Line 159-160; how can the bodyweight and genetic background of a dog be similar to humans?

There are several typo’s throughout the text. Some examples: line 55; dystrobrevin

line 246: 2’-O-methyl-phoshporothioneto.

Introduction: line 33; what kind of radical therapy are you referring?

References are lacking for lines 1-48. The one reference provided at line 50 is too late.

Rephrasing needed for sentences: Line 60-61, lines 73-75, line 86-88

Sentences 81-83 overlap with 119-121 & 123-124.

Author Response

Overall comment:

1.      The manuscript needs thorough editing of English language. Many sentences are incorrect due to incorrect use of English i.e. line 41 (use of noticed), line 47 patients have a de novo mutation.

Response:

The manuscript has been edited by a native of English again.

Major points:

1.      The review is not focused. Based on the title the review is about therapies which restore mutations in the DMD gene from diagnosis to therapy. A large part of the review deals however with ‘other non-related issues’ and clear structure is lacking. All information regarding therapies which target ‘secondary pathology’ is not relevant for this particular review. The author only refers to a few selected strategies, without really explaining. It does not add anything to the review. I would advice to delete it: Figure 1B and all text about this (section 5.1).

Response:

Thank the reviewer for the suggestion and advise. Actually, I only referred to a few selected strategies, without really explaining. Accordingly, I deleted Figure 1B and all text about this (section 5.1).

2.      The review is about targeting the DMD gene to restore dystrophin expression. The author only hints at the CRISPR/Cas9 developments in line 307. Explaining this should deserve a separate paragraph.

Response:

Thank the reviewer for the suggestion. I explained the CRISPR/Cas9 developments deserved a separate paragraph.

3.      The table 1 needs explanation.

Response:

I added the explanation.

4.      Some parts of the text are very confusing for non-expert readers. An example: the entire DMD gene does not fit in a viral vector (but only 4.9 kb as mentioned in line 195). Author start the paragraph however with … by introducing it to muscle cells using a full-length or functionally truncated DMD gene (cDNA) into a viral vector (line 191-192). Only 3 sentences later, the reader will read that what was just said (inserting a full-length cDNA) is in fact not possible. Rewrite this part and be more clear from the start onwards. Author refers to both mini and micro dystrophin. This difference should be explained. The part about gene therapy using viral vectors also lacks the more recent developments. There is no reference at all made to the work done so far in DMD patients with micro- and mini-dystrophin. In addition, it would be good for a reader to understand the proc and conc of each of the described therapeutic strategies. For each of the therapeutic strategies pictures explaining the technique should be provided like is done for exon skipping. (i.e. for this part; which mini, micro-dystrophins have been used in which clinical trial etc.).

Response:

Thank the reviewer for the appropriate comments. I greatly changed the constitution of the paragraphs and sentences of the applicable points (section 5.1)

5.      Also the order of the paragraphs would make more sense if it first deals with therapies which have conditional approval followed by those which are still in phase I/II.

Response:

Following the reviewer suggestion, the order of the paragraphs would make more sense if it first deals with therapies which have conditional approval followed by those which are still in phase I/II.

6.      The information provided about PTC124 is unclear for non-experts. There is no reference made in the text that PTC124 is the same as Ataluren and Translarna. Now sentence 220-221 make no link to PTC124 for readers who are not aware that this is the tradename. Also in sentence 218 it is mentioned that use was prevented due to initial trial outcomes. However, this was later changed. Please update the text.

Response:

According the reviewer’s comments, I have added references about PTC124 and the term was consisted. I have also updated the text.

7.      Line 229: it is indeed expected that depending on the mutation that will be corrected and the size and make-up of the synthesized dystrophin protein, there will be differences in therapeutic outcome. However, it is not expected that DMD patient will become asymptomatic. This is misleading information and should be removed. Especially since the best dystrophin levels restored so far was 0.9% of wildtype. We are very far from treating a patient such that the phenotype will be asymptomatic. Author should remove all references to making a DMD patient asymptomatic.

Response:

Thank the reviewer for the comment. I have changed the sentences.

8.      Figure 2: the reading frame which is shown does not match with the reading frame shown in the examples, and it will not be clear to non-experts whether this is the situation of A or B. Please adapt so that both matches.

Response:

 I have changed the figure and its legend.

9.      The focus of the paragraphs starting at line 252 is skewed. Thirteen lines of text are on the pre-clinical development, while only 4 lines are on the trials conducted in patients and the FDA filings. No reference on data obtained, patient numbers used, trial setup etc. is made.

Response:

I have added the clinical trials conducted in patients and the FDA filings and references on data obtained, patient numbers used, trial setup etc.

Minor points:

1.      Author consistently refers to muscle atrophy in DMD. In patients the calf show muscle hypertrophy. The rest of the skeletal muscles are wasted over time, but they are not atrophic.

Response:

Thank the reviewer for the comments. I have deleted the term of muscle atrophy.

2.      Line 136; mdx mice do develop heart failure from ~6 months of age onwards. The reference the author use is studying young mice, and does not study heart function by advanced techniques like MRI.

Response:

According the reviewer comments, I have newly cited the appropriate references concerning old mice and deleted the sentence about exercising study using young mdx mice with the referenced paper.

3.      Line 159-160; how can the bodyweight and genetic background of a dog be similar to humans?

Response:

As indicated by the reviewer, the sentence was confused by the readers. Therefore, I deleted the part.

4.      There are several typo’s throughout the text. Some examples: line 55; dystrobrevin

Response:

I have corrected the typos.

5.      line 246: 2’-O-methyl-phoshporothioneto.

Response:

I have corrected the typos.

6.      Introduction: line 33; what kind of radical therapy are you referring?

Response:

I made an error in writing and have corrected the phrase “So far, a radical therapy has not been developed.”

7.      References are lacking for lines 1-48. The one reference provided at line 50 is too late.

Response:

I added the references according to the reviewer comments.

8.      Rephrasing needed for sentences: Line 60-61, lines 73-75, line 86-88

Response:

I rephrased the sentences corresponding parts.

9.      Sentences 81-83 overlap with 119-121 & 123-124.

Response:

I deleted the sentences lines 119-121 & 123-124.

Round  2

Reviewer 1 Report

the authors have included nitric oxide only in passing and failing to mention the relevant references. What I was suggesting was to dedicate a paragraph at least to it

I am adding here a list of references as they cone when you simply put nitric die and duchenne in pubmed. There are many  the authors can choose from.

1: Timpani CA, Hayes A, Rybalka E. Therapeutic strategies to address neuronal nitric oxide synthase deficiency and the loss of nitric oxide bioavailability in  Duchenne Muscular Dystrophy. Orphanet J Rare Dis. 2017 May 25;12(1):100. doi: 10.1186/s13023-017-0652-y. Review. PubMed PMID: 28545481; PubMed Central PMCID: PMC5445371. 2: Miglietta D, De Palma C, Sciorati C, Vergani B, Pisa V, Villa A, Ongini E, Clementi E. Naproxcinod shows significant advantages over naproxen in the mdx model of Duchenne Muscular Dystrophy. Orphanet J Rare Dis. 2015 Aug 22;10:101. doi: 10.1186/s13023-015-0311-0. PubMed PMID: 26296873; PubMed Central PMCID: PMC4546261. 3: Cossu MV, Cattaneo D, Fucile S, Pellegrino P, Baldelli S, Cozzi V, Capetti A,  Clementi E. Combined isosorbide dinitrate and ibuprofen as a novel therapy for muscular dystrophies: evidence from Phase I studies in healthy volunteers. Drug Des Devel Ther. 2014 May 2;8:411-9. doi: 10.2147/DDDT.S58803. eCollection 2014. PubMed PMID: 24851040; PubMed Central PMCID: PMC4018313. 4: Wang G, Lu Q. A nitrate ester of sedative alkyl alcohol improves muscle function and structure in a murine model of Duchenne muscular dystrophy. Mol Pharm. 2013 Oct 7;10(10):3862-70. doi: 10.1021/mp400310r. Epub 2013 Aug 27. PubMed PMID: 23924275. 5: Ramachandran J, Schneider JS, Crassous PA, Zheng R, Gonzalez JP, Xie LH, Beuve A, Fraidenraich D, Peluffo RD. Nitric oxide signalling pathway in Duchenne muscular dystrophy mice: up-regulation of L-arginine transporters. Biochem J. 2013 Jan 1;449(1):133-42. doi: 10.1042/BJ20120787. PubMed PMID: 23009292; PubMed  Central PMCID: PMC4365916. 6: De Palma C, Clementi E. Nitric oxide in myogenesis and therapeutic muscle repair. Mol Neurobiol. 2012 Dec;46(3):682-92. doi: 10.1007/s12035-012-8311-8. Epub 2012 Jul 22. Review. PubMed PMID: 22821188. 7: D'Angelo MG, Gandossini S, Martinelli Boneschi F, Sciorati C, Bonato S, Brighina E, Comi GP, Turconi AC, Magri F, Stefanoni G, Brunelli S, Bresolin N, Cattaneo D, Clementi E. Nitric oxide donor and non steroidal anti inflammatory drugs as a therapy for muscular dystrophies: evidence from a safety study with pilot efficacy measures in adult dystrophic patients. Pharmacol Res. 2012 Apr;65(4):472-9. doi: 10.1016/j.phrs.2012.01.006. Epub 2012 Jan 25. PubMed PMID:  22306844. 8: Zhang Y, Duan D. Novel mini-dystrophin gene dual adeno-associated virus vectors restore neuronal nitric oxide synthase expression at the sarcolemma. Hum  Gene Ther. 2012 Jan;23(1):98-103. doi: 10.1089/hum.2011.131. Epub 2011 Oct 24. PubMed PMID: 21933029; PubMed Central PMCID: PMC3260444. 9: Thomas GD, Ye J, De Nardi C, Monopoli A, Ongini E, Victor RG. Treatment with a nitric oxide-donating NSAID alleviates functional muscle ischemia in the mouse model of Duchenne muscular dystrophy. PLoS One. 2012;7(11):e49350. doi: 10.1371/journal.pone.0049350. Epub 2012 Nov 5. PubMed PMID: 23139842; PubMed Central PMCID: PMC3489726. 10: Mizunoya W, Upadhaya R, Burczynski FJ, Wang G, Anderson JE. Nitric oxide donors improve prednisone effects on muscular dystrophy in the mdx mouse diaphragm. Am J Physiol Cell Physiol. 2011 May;300(5):C1065-77. doi: 10.1152/ajpcell.00482.2010. Epub 2011 Jan 26. PubMed PMID: 21270295. 11: Colussi C, Mozzetta C, Gurtner A, Illi B, Rosati J, Straino S, Ragone G, Pescatori M, Zaccagnini G, Antonini A, Minetti G, Martelli F, Piaggio G, Gallinari P, Steinkuhler C, Clementi E, Dell'Aversana C, Altucci L, Mai A, Capogrossi MC, Puri PL, Gaetano C. HDAC2 blockade by nitric oxide and histone deacetylase inhibitors reveals a common target in Duchenne muscular dystrophy treatment. Proc Natl Acad Sci U S A. 2008 Dec 9;105(49):19183-7. doi: 10.1073/pnas.0805514105. Epub 2008 Dec 1. Erratum in: Proc Natl Acad Sci U S A. 2009 Feb 3;106(5):1679. Steinkulher, Christian [corrected to Steinkuhler, Christian]. PubMed PMID: 19047631; PubMed Central PMCID: PMC2614736. 12: Brunelli S, Sciorati C, D'Antona G, Innocenzi A, Covarello D, Galvez BG, Perrotta C, Monopoli A, Sanvito F, Bottinelli R, Ongini E, Cossu G, Clementi E. Nitric oxide release combined with nonsteroidal antiinflammatory activity prevents muscular dystrophy pathology and enhances stem cell therapy. Proc Natl Acad Sci U S A. 2007 Jan 2;104(1):264-9. Epub 2006 Dec 20. PubMed PMID: 17182743; PubMed Central PMCID: PMC1765447. 13: Voisin V, Sébrié C, Matecki S, Yu H, Gillet B, Ramonatxo M, Israël M, De la Porte S. L-arginine improves dystrophic phenotype in mdx mice. Neurobiol Dis. 2005 Oct;20(1):123-30. PubMed PMID: 16137573. 14: Chaubourt E, Fossier P, Baux G, Leprince C, Israël M, De La Porte S. Nitric oxide and l-arginine cause an accumulation of utrophin at the sarcolemma: a possible compensation for dystrophin loss in Duchenne muscular dystrophy. Neurobiol Dis. 1999 Dec;6(6):499-507. PubMed PMID: 10600405.

Author Response

Reviewer #

The authors have included nitric oxide only in passing and failing to mention the relevant references. What I was suggesting was to dedicate a paragraph at least to it I am adding here a list of references as they cone when you simply put nitric die and duchenne in pubmed. There are many the authors can choose from.

Response:

Thank you for your detailed comment and introducing me to many references. I understand the importance of nNOS in the pathomechanism of DMD and as a therapeutic target. In the previous revised manuscript, the description of nNOS was not sufficient, as pointed out by the reviewer, although other mechanisms and related therapies were mentioned. However, this review article deals with mutation based-therapeutic strategies and another reviewer strongly suggested that the pathomechanisms and their related therapeutic strategies are not necessary/should not be included in the review. I agree with this comment; therefore, I have referenced the role of NO and cited the references in the text. However, as you have kindly suggested, I have included some sentences together with the references recommended.

51: Ramachandran J, Schneider JS, Crassous PA, Zheng R, Gonzalez JP, Xie LH, Beuve A, Fraidenraich D, Peluffo RD. Nitric oxide signaling pathway in Duchenne muscular dystrophy mice: up-regulation of L-arginine transporters. Biochem J. 2013 Jan 1;449(1):13342.

52: De Palma C, Clementi E. Nitric oxide in myogenesis and therapeutic muscle repair. Mol Neurobiol. 2012 Dec;46(3):682-92.

53: Thomas GD, Ye J, De Nardi C, Monopoli A, Ongini E, Victor RG. Treatment with a nitric oxide-donating NSAID alleviates functional muscle ischemia in the mouse model of Duchenne muscular dystrophy. PLoS One. 2012;7(11):e49350.

54: Mizunoya W, Upadhaya R, Burczynski FJ, Wang G, Anderson JE. Nitric oxide donors improve prednisone effects on muscular dystrophy in the mdx mouse diaphragm. Am J Physiol Cell Physiol. 2011 May;300(5):C1065-77.

55: Brunelli S, Sciorati C, D'Antona G, Innocenzi A, Covarello D, Galvez BG, Perrotta C, Monopoli A, Sanvito F, Bottinelli R, Ongini E, Cossu G, Clementi E. Nitric oxide release combined with nonsteroidal anti-inflammatory activity prevents muscular dystrophy pathology and enhances stem cell therapy. Proc Natl Acad Sci U S A. 2007 Jan 2;104(1):264-9.

Reviewer 2 Report

Since the last review, the author has made some major changes to the review article ‘‘Mutation-based Therapeutic Strategy for Duchenne Muscular Dystrophy: from genetic diagnosis to therapy’. As a result, it is a much more thorough and well written article.  In particular, the new figures are very useful.

However, there are some minor corrections that should be addressed:

-        Referencing: the author has vastly improved the use of referencing throughout the manuscript. However, there are formatting errors as they do not appear in the correct order (with numbers increasing throughout the text). For example, the reference numbering goes from 9 to 14. This should be corrected.

-        Wording: Some wording is a little vague. For example, it is still not clear what the difference is between ‘radical therapy’ and the ideas explained in the following section, such as exon skipping. Please specify.

-        Some sentences span 4 - 5+ lines and could be shortened for ease of reading

-        Title: for better English, the title should be ‘strategies’

Once the above points have been addressed I would be happy to accept this manuscript for publication.

Author Response

Reviewer #2

Since the last review, the author has made some major changes to the review article ‘‘Mutation-based Therapeutic Strategy for Duchenne Muscular Dystrophy: from genetic diagnosis to therapy’. As a result, it is a much more thorough and well written article.  In particular, the new figures are very useful. However, there are some minor corrections that should be addressed:

1.       Referencing: the author has vastly improved the use of referencing throughout the manuscript. However, there are formatting errors as they do not appear in the correct order (with numbers increasing throughout the text). For example, the reference numbering goes from 9 to 14. This should be corrected.

Response:

I apologize for the oversight and have accordingly corrected the chronological ordering of the references throughout the manuscript.

2.       Wording: Some wording is a little vague. For example, it is still not clear what the difference is between ‘radical therapy’ and the ideas explained in the following section, such as exon skipping. Please specify.

Response:

As commented by the reviewer, “radical therapy” was vague. With “radical,” I meant therapy that is aimed at complete recovery; therefore, I have changed it to “therapy aimed at complete recovery.”

3.       Some sentences span 4 - 5+ lines and could be shortened for ease of reading

Response:

I apologize for the oversight, and have accordingly shortened long sentences to improve readability.

4.       Title: for better English, the title should be ‘strategies’

Response:

       I have changed the title as recommended.

Reviewer 3 Report

Comments:

There are still multiple sentences which English is of bad quality.

Some examples: The authors consistently use only DMD when the refer to the gene. This should be adapted into ‘the DMD gene’. This occurs i.e. in lines 33, 48 and 49. Please adapt accordingly.

Line 60: should read: and its location within the DMD gene. Line: 445 is not correct English.

In the abstract the author mentions that the exon skip therapy could convert the phenotype in mild or asymptomatic. With current knowledge on clinical trials, much more potent compounds are needed for this. I am skeptical whether we will ever achieve the conversion into asymptomatic (‘theoretically” yes, however ‘practically’ not convinced), and suggest to tone this statement down, such that readers get a more realistic view.

I do not understand what the author means in table 1 with the last two columns in case of the DMD/BMD patients. It now reads that 20-50% of intact protein is produced in BMD patients. Why normal and abnormal for BMD and not detected for DMD while it still mentions a number; than it is detected in low level right (due to spontaneous exon skip)? 

Line 111 and 112: Following necrosis, muscle regeneration actively causes disease progression. …..and finally muscle fibers infiltrate fibrous and fatty tissues.  Are both incorrect. Muscle regeneration is not actively negatively influencing disease progression. Muscle fibers do not infiltrate other tissues, it is the other way around.

Figure 1 directly links sarcolemmal disruption and retardation of regeneration.

This is not correct. Chronic fibrosis and inflammation impair regeneration.

I do not understand what the authors wants to point out with line 119. This does not make sense at all. Line 119: Replace C57BL/10 with C57BL/10ScSnJ.

Line 120 + 128: gene names which refer to murine origin should be written with only the first letter in capital. Change DMD into Dmd.

Mdx should be written in italics; changes are needed throughout the manuscript.

The paragraphs on gene therapy should be more specific on the loading capacity of the individual vectors. For non-experts it is hard to understand why in some one uses micro-or mini dystrophin, while for other approached the full-length can be used. This should be already mentioned around line 172. Also a line on which domains are essential in the mini-micro dystrophin seems essential.

The reading frame shown in figure 2 does not match with the figures above. Please adapt the lower picture, such that it matches the example.

Line 173; micro or mini dystrophin is not ‘functionally truncated’  It is a truncated protein that is partially functional.

I do not understand line 422 from ‘together with age’ onwards.

I would recommend that the authors state in the paragraph on Eteplirsen the amount of dystrophin which has been achieved at study completion. (namely only 0.9% in one patient, with lower levels in the others).

The future aspects on therapeutic strategies paragraph is limited. What are the pros and coins, what are the difficulties which will be faced for each of the therapies? Etc.

Author Response

Reviewer #3

1.       There are still multiple sentences which English is of bad quality. Some examples: The authors consistently use only DMD when the refer to the gene. This should be adapted into ‘the DMD gene’. This occurs i.e. in lines 33, 48 and 49. Please adapt accordingly.

Response:

As recommended by the reviewer, I have changed instances of “DMD” to “the DMD gene” throughout the manuscript.

2.       Line 60: should read: and its location within the DMD gene. Line: 445 is not correct English.

Response:

I have revised lines 60 and 445 accordingly.

3.       In the abstract the author mentions that the exon skip therapy could convert the phenotype in mild or asymptomatic. With current knowledge on clinical trials, much more potent compounds are needed for this. I am skeptical whether we will ever achieve the conversion into asymptomatic (‘theoretically” yes, however ‘practically’ not convinced), and suggest to tone this statement down, such that readers get a more realistic view.

Response:

As requested by the reviewer, I have deleted the term “asymptomatic” from the abstract.

4.       I do not understand what the author means in table 1 with the last two columns in case of the DMD/BMD patients. It now reads that 20-50% of intact protein is produced in BMD patients. Why normal and abnormal for BMD and not detected for DMD while it still mentions a number; than it is detected in low level right (due to spontaneous exon skip)?

Response:

As Table 1 was not easy to understand, I have divided Table 1 into two tables, one for DNA analysis and one for protein assays. The last two columns mean that 20–50% of intact protein is produced or 20–100% of abnormal protein is produce in BMD patients.

5.       Line 111 and 112: Following necrosis, muscle regeneration actively causes disease progression. …..and finally muscle fibers infiltrate fibrous and fatty tissues. Are both incorrect. Muscle regeneration is not actively negatively influencing disease progression. Muscle fibers do not infiltrate other tissues, it is the other way around.

Response:

Thank you for your insightful comment. I have rewritten the sentences as follows,

“Following necrosis, muscle regeneration is retarded, which causes disease progression as regeneration cannot catch up with necrosis. Finally, fibrous and fatty tissues infiltrate the muscle fibers.“

6.       Figure 1 directly links sarcolemmal disruption and retardation of regeneration. This is not correct. Chronic fibrosis and inflammation impair regeneration.

Response:

I apologize for the oversight and have accordingly corrected Figure 1.

7.       I do not understand what the authors wants to point out with line 119. This does not make sense at all. Line 119: Replace C57BL/10 with C57BL/10ScSnJ.

Response:

I apologize for the confusion. I have thus deleted the sentences and replaced C57BL/10 with C57BL/10ScSnJ.

8.       Line 120 + 128: gene names which refer to murine origin should be written with only the first letter in capital. Change DMD into Dmd.

Response:

I have changed DMD to Dmd accordingly.

9.       Mdx should be written in italics; changes are needed throughout the manuscript.

Response:

As requested, I have made this change throughout the manuscript.

10.   The paragraphs on gene therapy should be more specific on the loading capacity of the individual vectors. For non-experts it is hard to understand why in some one uses micro-or mini dystrophin, while for other approached the full-length can be used. This should be already mentioned around line 172. Also a line on which domains are essential in the mini-micro dystrophin seems essential.

Response:

Thank your insightful suggestions. I have included a sentence regarding the loading capacity of individual vectors. Furthermore, I have prepared a new figure (Figure 2) that illustrates mini- and micro-dystrophin structure and its essential domains.

11.   The reading frame shown in figure 2 does not match with the figures above. Please adapt the lower picture, such that it matches the example.

Response:

Thank you for your comment. I have made the requested change, as indicated figure 3.

12.   Line 173; micro or mini dystrophin is not ‘functionally truncated’ It is a truncated protein that is partially functional.

Response:

I have rewritten the sentence accordingly.

13.   I do not understand line 422 from ‘together with age’ onwards.

Response:

I have deleted the sentence.

14.   I would recommend that the authors state in the paragraph on Eteplirsen the amount of dystrophin which has been achieved at study completion. (namely only 0.9% in one patient, with lower levels in the others).

Response:

Thank you for the suggestion. I have included a sentence regarding the amount of dystrophin achieved at study completion.

15.   The future aspects on therapeutic strategies paragraph is limited. What are the pros and coins, what are the difficulties which will be faced for each of the therapies? Etc.

Response:

I am grateful for your suggestion. I have accordingly revised the paragraph by including the pros and cons and the difficulties that will be faced by each therapy strategy as follows,

Revised paragraph

Despite being a single gene disorder, various therapeutic approaches have been developed against the background of the complex pathological mechanism of DMD. The most fundamental treatment for DMD is considered supplementation or recovery of dystrophin. In this regard, the use of viral vectors, exon-skipping, or readthrough therapies as described in this review are desirable. However, each treatment approach also faces various difficulties. Although the use of viral vectors can be widely adapted regardless of the gene mutation, there is a limit to the size of genes that can be incorporated while also ensuring safety. Meanwhile, exon-skipping therapy is tailor-made based on the genetic mutation; however, this limits the number of target patients, and there are other problems regarding introduction efficiency and symptom improvement. For readthrough therapy, the number of subjects that can be treated is also very limited. Moreover, in any treatment, the type and number of target organs (skeletal muscle, myocardium and smooth muscle, and nervous system) of DMD patients are both varied and high, which is another hurdle for these therapeutic strategies to overcome. Although these issues may be resolved in the future, it is necessary to combine several drug therapies.